# GS-2: A Novel Broad-Spectrum Agent for Environmental Microbial Control

**DOI:** 10.3390/biom12091293

**Published:** 2022-09-13

**Authors:** Alyce J. Mayfosh, Zoe I. Day, Nathan B. Unsworth, Chun-Qiang Liu, Ruchi Gupta, Soraya Haynes, Rebecca Abraham, Sam Abraham, Zo L. Shaw, Sumeet Walia, Aaron Elbourne, Mark D. Hulett, Thomas F. Rau

**Affiliations:** 1Ten Carbon Chemistry, PO Box 4317, Hawker, ACT 2614, Australia; 2Department of Biochemistry and Chemistry, School of Agriculture, Biomedicine and Environment, La Trobe Institute for Molecular Science, La Trobe University, Plenty Rd, Bundoora, VIC 3086, Australia; 3Defence Science and Technology Group, 506 Lorimer Street, Fishermans Bend, VIC 3207, Australia; 4Antimicrobial Resistance and Infectious Diseases Laboratory, Harry Butler Institute, Murdoch University, Murdoch, WA 6150, Australia; 5School of Engineering, RMIT University, Melbourne, VIC 3001, Australia; 6School of Science, RMIT University, Melbourne, VIC 3001, Australia

**Keywords:** antimicrobial, sanitizer, disinfectant, antiviral, antifungal, antibacterial, thymol, COVID-19, fatty acid, capric acid

## Abstract

The environmental control of microbial pathogens currently relies on compounds that do not exert long-lasting activity on surfaces, are impaired by soil, and contribute to the growing problem of antimicrobial resistance. This study presents the scientific development and characterization of GS-2, a novel, water-soluble ammonium carboxylate salt of capric acid and L-arginine that demonstrates activity against a range of bacteria (particularly Gram-negative bacteria), fungi, and viruses. In real-world surface testing, GS-2 was more effective than a benzalkonium chloride disinfectant at reducing the bacterial load on common touch-point surfaces in a high-traffic building (average 1.6 vs. 32.6 CFUs recovered from surfaces 90 min after application, respectively). Toxicology testing in rats confirmed GS-2 ingredients were rapidly cleared and posed no toxicities to humans or animals. To enhance the time-kill against Gram-positive bacteria, GS-2 was compounded at a specific ratio with a naturally occurring monoterpenoid, thymol, to produce a water-based antimicrobial solution. This GS-2 with thymol formulation could generate a bactericidal effect after five minutes of exposure and a viricidal effect after 10 min of exposure. Further testing of the GS-2 and thymol combination on glass slides demonstrated that the compound retained bactericidal activity for up to 60 days. Based on these results, GS-2 and GS-2 with thymol represent a novel antimicrobial solution that may have significant utility in the long-term reduction of environmental microbial pathogens in a variety of settings.

## 1. Introduction

To date, hospital-acquired infections account for an estimated 1.7 million infections and 99,000 deaths annually [1]. While many pathogens originate from patients, an estimated 20–40% of nosocomial infections are caused by cross-contamination from the hands of healthcare workers [2]. This cross-contamination likely results from healthcare personnel touching contaminated surfaces that are sanitized with agents that lack adequate antimicrobial activity [3]. Indeed, the current SARS-CoV-2 pandemic has drawn renewed attention to the environmental control of microorganisms and the role contaminated surfaces play in the spread of infectious diseases. One study demonstrated that SARS-CoV-2 can survive at room temperature on hard surfaces for at least 28 days [4] and on hands for approximately nine hours [5]. Of particular importance is the control of SARS-CoV-2 contamination in hospitals.

There is a growing body of evidence that current disinfection solutions may be ineffective at reducing hard surface contamination and nosocomial infection in healthcare environments. Despite strict disinfectant regimens, common hospital pathogens, such as *Clostridium difficile*, *Acinetobacter baumanni*, methicillin-resistant *Staphylococcus aureus* (MRSA), and vancomycin-resistant *Enterococcus* (VRE), frequently colonize surfaces for extended periods, leading to direct patient infection [6,7,8,9]. Currently, many disinfectants utilized across multiple healthcare and industrial sectors rely on quaternary ammonium compounds (quats) as their active ingredients [10]. Quats have numerous advantages: they are inexpensive and have broad activity against bacteria, fungi, and enveloped viruses. However, they have poor activity against *C. difficile* spores, *Mycobacterium tuberculosis*, and non-enveloped viruses, and their activity is adversely affected by organic matter or soil [11,12]. Furthermore, questions have recently been raised as to the safety of quats, where quats in rodents have been shown to affect neural tube formation, reproduction, and respiration and are genotoxic to cells, as well as ecotoxic to aquatic life [13,14,15,16,17,18]. In humans, continuous exposure has been demonstrated to produce contact dermatitis and, in certain cases, induce asthma [19,20].

Another significant issue associated with quats is the induction of cross-resistance to antibiotics, where exposure to the common quat benzalkonium chloride (BAC) was shown to increase the resistance of MRSA to beta-lactams [21] and increase the resistance of enterohemorrhagic *Escherichia coli* to six different, commonly used antibiotics [22]. In addition to the development of cross-resistance, the use of quats also produces bacterial resistance to the disinfectant itself through the upregulation of efflux pumps, decreased porins related to BAC transport, and increased expression of spermidine and *pmrB* resulting in a reduction in the surface membrane negative charge (since BAC is cationic) [23]. Based on the current rate of nosocomial infection coupled with the safety and performance issues associated with quats, there is a growing need to develop safe alternatives that are capable of effectively controlling pathogenic microorganisms within the environment.

Antimicrobial fatty acids represent a known but relatively underutilized resource in the fight against infective organisms. A wide range of fatty acids has been found to have broad-spectrum antibacterial activity [24]. There are several advantages to utilizing fatty acids in the environmental control of microbes: they do not appear to develop resistance [25]; they are a naturally occurring component in the human innate immune response to pathogens, particularly on skin [26,27]; they have extremely low toxicity risk associated with their use (LD_50_ in rats fed capric acid exceeded 10 g/kg [28]); and they rapidly undergo biodegradation in the body and are utilized as an energy source. Capric acid (also known as decanoic acid), found naturally in coconut oil (~10%), palm kernel oil, and goat milk, is rapidly metabolized to ketones through beta-oxidation and utilized in the mitochondria for energy [29].

While there is renewed interest in fatty acids as effective antibacterial agents, their stability and solubility have seen limited development. Physically, fatty acids are almost completely water-insoluble and must be combined with solubilizing agents, dissolved in organic solvents, or chemically modified (e.g., methyl esterification) to be utilized in practical applications. However, these modifications can reduce the efficacy of fatty acids when compared to their unmodified forms [30]. Capric acid, for example, is highly soluble in alcohol, ether, DMSO, and acetone but relatively insoluble in water [31]. Overcoming this challenge of presenting fatty acids in a safe and soluble form would present a novel agent for use as an environmental antimicrobial and sanitizer.

The present study details the development of a novel antimicrobial agent containing a naturally occurring 10-carbon fatty acid, capric acid, combined with L-arginine to produce a water-soluble ammonium carboxylate salt. The further addition of a 10-carbon monoterpenoid, thymol, provides fast-acting, broad-spectrum antimicrobial and sanitizing activity against multidrug-resistant hospital pathogens in both in vitro and real-world settings.

## 2. Materials and Methods

### 2.1. Nuclear Magnetic Resonance (NMR)

An aliquot (250 μL) of GS-2 was diluted with 20 mL MilliQ water, adjusted to pH 4, and extracted 3 times with 40 mL heptane. The aqueous fraction was freeze-dried and reconstituted with D2O for ^1^H and ^13^C analysis. The combined heptane fractions were dried under vacuum and taken into CDCl_3_ for ^1^H and ^13^C analysis.

### 2.2. Gas Chromatorgraphy Mass Spectrometry (GCMS)

GS-2 was analyzed as a mixture, followed by analyses of both the aqueous and heptane-extracted fractions. All detected peaks were matched against the National Institute of Standards and Technology (NIST) database for annotation. Following whole formulation analysis, 250 μL of GS-2 was diluted with 20 mL MilliQ water and extracted 3 times with 40 mL heptane (Sigma-Aldrich, St. Louis, MO, USA). The aqueous fraction was freeze-dried; derivatized with n, o-bis (trimethylsilyl) trifluoroacetamide; and resuspended in methanol (Sigma-Aldrich, St. Louis, MO, USA) for trace analysis. The combined heptane fractions were dried under vacuum, resuspended in n-hexane, and analyzed separately.

### 2.3. Determining Activity of GS-2 against Bacterial and Fungal Pathogens

The minimum inhibitory concentration (MIC), minimum bactericidal concentration (MBC), and minimum fungicidal concentration (MFC) of both GS-2 and GS-2 with thymol were determined using a broth dilution method under Clinical Laboratory Standards Institute (CLSI) M26-A guidelines [32]. *C. difficile* (American Type Culture Collection (ATCC), Manassas, VA, USA) was cultured on BHI-YHT agar for 48 h prior to testing. *Candida auris* (Centers for Disease Control and Prevention (CDC) AR Bank 0381, Atlanta, GA, USA) was cultured on Sabouraud dextrose agar for 24 h prior to testing. *Clostridium butyricum* (ATCC 19398), *S. aureus* (ATCC 6538), *Pseudomonas*
*aeruginosa* (ATCC 15442), and *Klebsiella pneumoniae* (ATCC 700603) were cultured on tryptic soy agar (TSA) with 5% sheep’s blood for 24 h prior to testing. *Bacillus atrophaeus* (ATCC 9372) and *E. coli* (ATCC 25922) were cultured on TSA plates for 24 h prior to testing.

Organism suspensions were prepared at 1 × 10^5^ CFU/mL in appropriate culture media as follows: *S. aureus*, *E. coli*, *K. pneumoniae*, and *P. aeruginosa* in Luria–Bertani broth; *C. auris* in Sabouraud dextrose broth; *C. butyricum* in reinforced clostridium medium; *B. atrophaeus* in Mueller–Hinton broth; *C. difficile* in brain–heart infusion broth. GS-2 and brain–heart infusion broth were prepared 24 h prior to testing to allow pre-reduction in an anaerobic chamber overnight for the growth of *C. difficile*.

Serial dilutions (1:1) of both GS-2 and GS-2 with thymol were prepared in culture media specified above in triplicate for each organism in a 96-well plate. Bacterial and fungal cultures (100 µL) were then added to the diluted GS-2 tests. Plates were then incubated for 24–48 h, either at 30 °C (*C. auris*), at 36 °C in anaerobic conditions (*C. butyricum* and *C. difficile*), or 36 °C in aerobic conditions (*S. aureus*, *B. atrophaeus*, *E. coli*, *K. pneumoniae*, and *P. aeruginosa*). MIC, MBC, and MFC were determined via visual assessment.

### 2.4. Determining Activity of GS-2 against Clinically Isolated Bacterial Pathogens

GS-2 activity was tested against four clinically isolated pathogens: MRSA, *Streptococcus pyogenes*, and the carbapenem-resistant enterobacteriaceae (CRE) *E. coli* and *K. pneumoniae* (samples generously donated from Kalispell Regional Medical Center, Montana) using a 96-well microplate format under CLSI M26-A guidelines [32]. A serial dilution in Oxoid Sensitest medium of GS-2 was combined with bacteria to determine the MIC and MBC of GS-2 after 24 h of incubation. MIC was determined by visual assessment with no turbidity. MBC was determined by combining technical replicates and centrifuging to pellet bacteria. The supernatant was removed, and the pellet was rehydrated with 1 mL sterile saline. One hundred microliters of this solution was plated onto 5% sheep blood agar (SBA) plates and incubated for 24 h at 37 °C. MBC was the concentration of GS-2 that produced a 99.99% reduction from the average starting inoculum. This value was calculated by a manual dilution series of the inoculum on 5% SBA plates (the starting inoculum was diluted, plated, incubated for 24 h, and counted to obtain an accurate starting concentration for the percent reduction calculation). For growth and contamination controls, media-only and water controls were set up on the 96-well plate and processed consistently with experimental samples to ensure sterility and quality control. Positive growth control wells containing bacteria, media, and sterile water were diluted, plated, and counted to determine the amount of growth from the starting inoculum and to ensure robust growth against the test organism.

### 2.5. Testing GS-2 and GS-2 with Thymol under Regulatory Disinfectant Standards

MRSA and *E. coli* (clinical isolates) were tested against GS-2 and GS-2 with thymol according to the Therapeutic Goods Administration (TGA) Option C household/commercial-grade disinfectant guidelines [33]. Briefly, bacteria were grown to log-phase in Oxoid Sensitest broth at 37 °C under constant rotation for 6 h. Three milliliters of either 15 mg/mL GS-2, DMSO, 5 mg/mL thymol in DMSO, 30 mg/mL GS-2 in DMSO, or 30 mg/mL GS-2 with 5 mg/mL thymol was combined with 9 × 10^8^ CFU/mL bacteria in 1 mL and incubated for 8 min at room temperature. After 8 min, samples were vortexed, and 20 µL was placed in 5 mL Oxoid Sensitest medium and incubated for 48 h. A pass result required 2 or more of the 5 tubes to be clear (indicating no growth) after 48 h.

### 2.6. Antibacterial Time-kill Studies with GS-2 against Potential Bioterrorism Pathogen Surrogates

Cultures of *Burkholderia thailandensis* (ATCC 700388), a surrogate for *Burkholderia pseudomallei,* and *Bacillus anthracis* Sterne 34F2 (Fort Dodge Australia Pty Ltd., Colorado Serum Company, Denver, CO, USA) were prepared in nutrient broth and incubated overnight at 37 °C with shaking. Cultures were diluted to an OD_600_ equivalent of 2 × 10^6^ CFU/mL in fresh broth and incubated further for 1 h with shaking to obtain a log-phase culture. A final bacterial inoculum of 1 × 10^6^ CFU/mL in 0.5 mL of the log-phase culture was added to 0.5 mL of broth containing two-fold-concentrated GS-2 (final concentration 4.76 mg/mL), ciprofloxacin (final concentration 0.01 mg/mL), or meropenem (final concentration 0.10 mg/mL; Sigma-Aldrich, St. Louis, MO, USA). Cultures were incubated for 1 h, 3 h, 5 h, and 24 h (*B. thailandensis*) or 2 h, 4 h, 6 h, and 24 h (*B. anthracis*). Following each time-point, samples were washed in PBS, serially diluted, and plated onto nutrient agar plates. The plates were incubated at 37 °C for 24–48 h, and CFU/mL was calculated to determine time-kill kinetics.

### 2.7. Determining Antiviral Activity of GS-2 and GS-2 with Thymol

GS-2 was tested against Murine Hepatitis Virus Strain 1 (MHV-1; ATCC/VR-261, the severe acute respiratory syndrome-associated coronavirus (SARS-CoV) surrogate), Herpes Simplex Virus-1 (HSV-1; ATCC/VR-733), and poliovirus (ATCC/VR-192) according to TMCV 006 ASTM International Standard E1053 [34]. Briefly, viral suspensions were prepared in 5% FBS, spread over the surface of a glass petri dish, and allowed to dry. Two milliliters of GS-2 (15 mg/mL) or GS-2 (30 mg/mL) with thymol (5 mg/mL) was then applied to the viral film and incubated for 10 min at RT. Following incubation, viruses were harvested from test and control slides with a cell scraper, and GS-2 or GS-2 with thymol was neutralized with 2% FBS in minimum essential medium (MEM). Separately, 2 mL GS-2 or GS-2 with thymol was also combined with 2% FBS in MM and then either combined with viral suspension to ensure neutralization (neutralization control) or set aside for the following step (cytotoxicity control).

The neutralized harvested test suspensions, neutralization control, and cytotoxicity control were serially diluted and plated in quadruplicate onto a 96-well plate containing a monolayer of either A9 cells (ATCC/CCL-1.4; for MHV-1) or Vero cells (ATCC/CCL-81; for HSV-1 and poliovirus). Four wells containing cells only served as controls. Media was added to all wells and incubated for 7–10 days. Then, each well was examined for the presence of cytopathic effects of infection, such as cell rounding, sloughing, and monolayer degradation. Cytotoxicity control wells were examined for damage caused by the test product. The Reed and Muench LD_50_ method was used to determine the viral titer endpoint.

### 2.8. GS-2 Surface Sanitizer Trial

Two commercial high-rise buildings with near-identical layouts (built in the same year within 0.25 km of each other and monitored for daily traffic in the building with no significant differences in work population over the two-week test period) were sanitized for two weeks. Building A was sanitized with GS-2 at 15 mg/mL (1.5%), and building B was sanitized with a hospital-grade disinfectant containing 0.42% BAC according to product label instructions. After two weeks of treatment, swabs were taken in both buildings of five high-touch-point surfaces: loading dock bathroom door handles, office bathroom door handles for one level, office door handles into the loading dock, elevator buttons, and elevator touch screens (identical in both buildings). Swabs were taken 90 min before the morning treatment, 90 min after the morning treatment, and again at 4 h after treatment. Swabs were then cultured on 5% sheep blood agar (SBA) plates (Thermo Fisher Scientific, Scoresby, Australia) for 24 h at 37 °C, and the number of colonies was counted.

### 2.9. Toxicology Studies with GS-2

All animal handling and treatment was approved by the University of Melbourne Animal Ethics Committee. All animals were given free access to food, water, and enrichment. They were co-housed for a 72 h acclimation period prior to the beginning of experiments and then housed singly during the dosing protocol. Eight Sprague-Dawley rats were given a subcutaneous bolus dose of GS-2 at 10 mL/kg under the skin between the shoulder blades. Tail vein blood was drawn, processed, and assayed for capric acid at 1 h, 4 h, 8 h, and 24 h via LC-MS. During this period, the rats were observed for any adverse events or abnormal behavior. At 24 h, the animals were euthanized.

### 2.10. Hospital-Grade Disinfectant Testing EN13727

The GS-2 and thymol formulation (15.0 mg/mL GS-2 with 7.5 mg/mL thymol) was tested for activity against *S. aureus*, *Enterococcus hirae* (*E. hirae*; ATCC 10541), and *P. aeruginosa* according to European Standard (EN) 13727:2012 + A2:2015 [35]: hospital-grade disinfectant with 5 min exposure under dirty conditions. Briefly, bacterial cultures were prepared and diluted to 1.5–5 × 10^8^ CFU/mL. Each bacterial culture (1 mL) was combined with 1 mL interfering substance (3 g/L BSA, 3 mL/L erythrocytes) (THermo Fisher, Scoresby, Australia) and incubated at 20 °C for 2 min. Then, 8 mL GS-2 and thymol formulation was added and incubated for 5 min. Following incubation, 1 mL was transferred to a tube containing 8 mL neutralizer (polysorbate 80 (30 g/L) lecithin (3 g/L)) (Sigma-Aldrich, St Louis, MO, USA) and 1 mL water, mixed, incubated at 20 °C for 10 s, inoculated onto TSA plates in duplicate, and incubated overnight at 37 °C. Plates containing no colonies (i.e., ≥5-log reduction in growth) were considered a pass. Plates displaying one colony or more were considered a fail.

### 2.11. Testing Residual Surface Activity of GS-2 with Thymol

Sterile glass slides were coated with either 200 μL GS-2 with thymol or 200 μL sterile water and allowed to dry. On days 0, 15, 30, and 60, slides were inoculated with 1 × 10^6^ CFUs MRSA or *P. aeruginosa* and incubated for 60 min, uncovered, at room temperature. Viable bacteria were then extracted from the slides, diluted, plated onto 5% SBA plates, incubated overnight, and colonies were counted. Enumeration was performed as per NCCLS/CLSI guideline M26-A.

### 2.12. Statistics

Statistical analyses (Student’s unpaired two-tailed t-test) were performed with GraphPad Prism 9.0 software (GraphPad Software, San Diego, CA, USA).

## 3. Results

### 3.1. GS-2 Development and Characterization

Capric acid is a 10-carbon saturated fatty acid with known antibacterial and antifungal properties [36,37,38]. However, esterifying fatty acids to solubilize them has been shown to reduce activity [24,30]. Our initial goal was to discover a method for solubilizing the fatty acid without compromising the native structure. We also needed to solubilize the fatty acid without using toxic solvents to retain the generally regarded as safe and effective (GRASE) nature of the product. To achieve this, we screened a number of GRASE compounds under varying conditions to arrive at the ammonium carboxylate salt of capric acid and L-arginine.

Capric acid is typically solid at room temperature (melting point 31.4 °C) with water solubility at 0.15 mg/mL at 20 °C with a pH 7.0 (as described previously) [39]. However, once compounded with L-arginine under specific conditions, the solubility of capric acid in water at 20 °C exceeds 150 mg/mL (1000-fold increase). Based on the known chemistry and synthesis conditions, upon compounding, the basic amine group on L-arginine deprotonates the carboxylic group in capric acid to form a highly stable, unreactive ammonium carboxylate salt (Figure 1). This combination of capric acid and L-arginine in water is given the name GS-2.

To further characterize the formulation and to confirm that no other products were produced upon compounding, GS-2 was separated into an aqueous fraction and a heptane fraction and subjected to ^1^H and ^13^C NMR. Spectra detected in the aqueous and heptane fractions were consistent with the number of protons and carbons present in L-arginine (Appendix A) and capric acid (Appendix A), respectively, as well as consistent with published work [40,41,42,43].

A GCMS analysis of the aqueous and heptane fractions of GS-2 was also performed to detect any low concentrations of secondary products. As observed with NMR, only L-arginine (or the thermal decomposition structure of arginine) was detected in the aqueous fraction, and only capric acid was detected in the heptane fraction (Appendix A). These results suggest that there was no evidence of byproduct formation, confirming that capric acid and arginine remained non-covalently bound in solution and that no new compounds were formed upon combining.

After compounding, the pH of the final solution of GS-2 at room temperature was 8.0–8.5, which appeared to be mediated by the basic buffering capacity of L-arginine. Acidification of GS-2 to pH 5 using 1 M HCl resulted in the disassembly of bonding between capric acid and L-arginine, with capric acid falling completely out of the solution. However, at pH 7, the solution of capric acid and L-arginine remained fully soluble with no evidence of turbidity. The final compounding of GS-2 was optimized with capric acid at 150 mg/mL with L-arginine in water. Subsequent testing showed that capric acid could be increased up to 300 mg/mL in the presence of L-arginine and maintain solubility. However, this increased the synthesis time from 2 h to over 48 h and, therefore, was not pursued.

### 3.2. Antibacterial and Antifungal Activity of GS-2

After compounding and characterizing its physical properties, the antimicrobial activity of GS-2 was tested against common bacterial pathogens: *S. aureus*, *B. atrophaeus* (surrogate for *B. anthracis*), *C. butyricum* (surrogate for *Clostridium botulinum*), *C. difficile* (vegetative and endospores), *E. coli*, *K. pneumoniae*, and *P. aeruginosa*, as well as the fungal pathogen *C. auris*. The Gram-positive pathogens *S. aureus,*
*B. atropheaus, C. butyricum*, and *C. difficile* were most sensitive to GS-2, with MICs below 1 mg/mL and MBCs below 1.2 mg/mL (Table 1). The Gram-negative rods (GNR) *E. coli, K. pneumoniae*, and *P. aeruginosa* displayed significantly higher MICs (3.75–7.5 mg/mL) and MBCs (3.75–10 mg/mL) (Table 1), suggesting GS-2 was more potent against Gram-positive than Gram-negative pathogens. The MIC/MFC value observed against *C. auris* was similar to Gram-positives (Table 1), suggesting that GS-2 was also potent against fungal species. Testing with capric acid in DMSO against *E. coli* and *K. pneumoniae* did not produce an MIC or MBC, indicating that the combination of capric acid and L-arginine produced an enhanced antibacterial effect.

The antibacterial activity of GS-2 was then tested against clinical isolates of four common bacterial pathogens: MRSA, *S. pyogenes* (Group A β-hemolytic strep), and CRE species *K. pneumoniae* and *E. coli*. The Gram-positive cocci (GPC) MRSA and *S. pyogenes* were most sensitive to GS-2, with *S. pyogenes* demonstrating the greatest sensitivity at an effective MBC of 0.78 mg/mL, whereas the Gram-negative rods (GNR) *E. coli* and *K. pneumoniae* displayed significantly higher MBCs (7.04 mg/mL and 8.04 mg/mL, respectively) (Table 2). As a control, the use of L-arginine alone in water at 50 mg/mL did not produce an MIC or MBC in any of the bacteria tested. Capric acid in water at solubility (0.15 mg/mL) also failed to achieve an MIC or MBC in any of the four organisms tested here, again suggesting that the combination of capric acid and L-arginine conferred greater activity than the fatty acid alone.

A criterion for disinfectants is a rapid time-kill. To determine the suitability of GS-2 as a disinfectant, we performed TGA disinfectant testing, standards relevant to Australia that test a disinfectant’s activity against high concentrations of bacteria within 8 min. GS-2 passed this testing against *E. coli* but failed against MRSA (Table 3), suggesting that GS-2 could rapidly kill Gram-negative bacteria but required a longer exposure time to kill Gram-positive bacteria.

The activity of GS-2 was then tested against pathogens that cause significant disease and pose a high risk to be weaponized in circumstances such as bioterrorism. GS-2 was tested against *B. thailandensis,* a surrogate for *B. pseudomallei* that causes melioidosis, and *B. anthracis*, which causes anthrax. The treatment of *B.*
*thailandensis* with GS-2 at 4.76 mg/mL resulted in a six-log decrease in growth and complete growth inhibition after 1 h and persisted 5 h after treatment (Figure 2A). Interestingly, GS-2 activity was superior to the antibiotic meropenem, the current first-line therapy against melioidosis, highlighting that the difference in effect between GS-2 was the time-kill effect and the antibiotic effects of meropenem. However, after 24 h, growth slightly recovered, with an average of 200 CFU/mL detected in both GS-2- and meropenem-treated samples. Testing of GS-2 against *B. anthracis* vegetative Sterne cells showed a rapid (four-log) decrease in growth in the first 2 h (Figure 2B). In contrast, treatment with ciprofloxacin, the first-line therapy for anthrax, resulted in a three-log reduction in *B. anthracis* vegetative cells.

### 3.3. GS-2 Exhibits Antiviral Activity

For GS-2 to be considered an appropriate broad-spectrum antimicrobial, it must also have activity against viruses. To determine its antiviral activity, GS-2 was tested against coronavirus MHV-1, a SARS-CoV-2 surrogate. GS-2 achieved a 4.92-log reduction in viral titer after 10 min of exposure against MHV-1 (Table 4). This showed that GS-2 was effective against a coronavirus, supporting its use in surface decontamination against SARS-CoV-2.

### 3.4. GS-2 Performs As an Effective Surface Sanitizer in Real-World Settings

Based on the antibacterial, antifungal, and antiviral activity of GS-2, the efficacy of GS-2 on surfaces was further tested. To test the effect of GS-2 in a real-world environment, two comparative commercial buildings were sanitized for two weeks with either GS-2 (15 mg/mL) or with a commercially available hospital-grade disinfectant containing 0.42% BAC according to label instructions. After two weeks (on day 15), five high-traffic touch-points in each building were swabbed and cultured before sanitizing on day 15, 90 min after sanitizing, and 4 h after sanitizing (Figure 3A). The first swab on day 15 before final sanitizing demonstrated that the building sanitized with 0.42% BAC for the previous 14 days had, on average, 5.6-fold more CFUs on touch-points than the building sanitized with GS-2, although the differences were not statistically significant (*p* = 0.18; Figure 3B). Ninety minutes after sanitizing on day 15, surfaces sanitized with 0.42% BAC hospital disinfectant had, on average, 20.3 times more CFUs than the GS-2-sanitized surfaces (*p* = 0.0087; Figure 3C). GS-2 activity appeared to persist longer than the BAC-containing disinfectant, and 4 h after sanitizing, there was a 7.9-fold reduction in CFUs recovered from GS-2-sanitized surfaces compared to BAC-treated surfaces (*p* = 0.17; Figure 3D). The authors acknowledge that these results are not wholly conclusive; there are variables in a ‘real-world’ study that could potentially bias these results, such as traffic through a building on a particular day and hand hygiene between people in buildings, and this is a single study between two buildings. However, the findings demonstrate that GS-2, under the tested conditions, provided antimicrobial activity at or better than a currently approved hospital-grade disinfectant.

### 3.5. GS-2 Poses No Toxic Effects and Is Rapidly Cleared In Vivo

Whilst capric acid is naturally found in foods such as coconut and palm kernel oil and L-arginine is a conditionally essential amino acid, we sought to confirm that there were no toxic effects of GS-2. Sprague-Dawley Rats were systemically challenged with GS-2 at 154.2 mg/mL at 10 mL of this solution per kg. The peak serum concentration of GS-2 (T_max_) was reached after 1.3 h, and capric acid was almost completely cleared from the rats after 24 h (Figure 4). We did not observe any abnormal behavior or any overt toxicity, even at this systemic dose, which very likely exceeded any potential environmental exposure. Given the fact that capric acid and L-arginine are commonly found in the diets of mammals, the lack of toxicity associated with high systemic exposure was not wholly unexpected.

### 3.6. The Addition of Thymol Improves Antimicrobial Activity of GS-2

Data to this point demonstrate that GS-2 provided potent, broad-range antimicrobial activity. However, our findings clearly indicated that GS-2 required longer exposure times to show activity against Gram-positive bacteria, such as *S. aureus* (when compared to the Gram-negative rods), and thus, was not suitable as a disinfectant due to the longer exposure times required. To address this, we tested several naturally occurring GRASE compounds that have previously been reported as having significant antimicrobial activity. After extensive screening, we found that GS-2 formed a stable solution with thymol. Thymol is a naturally occurring 10-carbon monoterpene phenol derived from the plant *Thymus vulgaris* that exhibits antibacterial properties.

To test the antimicrobial activity of this GS-2 and thymol formulation, this formulation, as well as GS-2 alone and thymol alone, was tested with a short exposure time of 8 min to represent fast activity. Both 5.0 mg/mL thymol in DMSO and 30 mg/mL capric acid in DMSO were not able to prevent bacterial growth of MRSA or CRE with *E. coli* in this short exposure time (Table 5). However, the combination of thymol (5.0 mg/mL) and GS-2 (30 mg/mL) was successful in completely inhibiting the growth of both MRSA and *E. coli* within 8 min (Table 5).

Testing of GS-2 and thymol against *S. aureus*, *E. coli*, *K. pneumoniae*, and *P. aeruginosa* resulted in 2–4-fold lower MICs and MBCs (Table 6) when compared to GS-2 alone (Table 1). The largest differences observed in the MICs and MBCs were against *E. coli*, *K. pneumoniae*, and *P. aeruginosa*, indicating that the addition of thymol improved activity against Gram-negative pathogens when exposed for 24 h.

To confirm the viricidal activity of this new formulation, GS-2 with thymol was tested against HSV-1, poliovirus, and MHV-1 with a 10 min exposure time using the TMCV 006 ASTM E1053 standard [34]. Testing of GS-2 with thymol against HSV-1, poliovirus, and MHV-1 produced > 6.5-log_10_, 3.0-log_10_, and 5.42-log_10_ reductions of viral growth, respectively (Table 7). Interestingly, the GS-2 with thymol formulation demonstrated activity against the non-enveloped virus of poliovirus, a property commonly lacking from current surface antimicrobials and disinfectants [11,12]. Overall, GS-2 with thymol demonstrated broad antibacterial and antiviral activity within 10 min of exposure.

After further testing, we refined the ratio of GS-2 to thymol to optimize the antibacterial activity. Based on our testing, we arrived a ratio of 2:1 (GS-2 : thymol) that provided optimal activity against both GNRs and Gram-positive cocci (GPC). As previously mentioned, GS-2 alone had substantial, rapid activity against GNRs; however, the formulation of ratios that moved away from the 2:1 ratio significantly extended the time required to kill GPCs. GS-2 at 15 mg/mL and thymol at 7.5 mg/mL successfully passed hospital-grade disinfectant testing using the EN13727 standard at 5 min of exposure under dirty conditions, achieving a >five-log reduction of growth for *S. aureus*, *E. hirae*, and *P. aeruginosa* (Table 8). Interestingly, a ratio of 6:1 (GS-2 to thymol) did not successfully pass this EN13727 testing, which suggested that the ratio of actives in the formulation is critical.

### 3.7. GS-2 with Thymol Remains Active on Surfaces

Based on the results of the building trial, it was hypothesized that GS-2 and the GS-2 with thymol formulation could bind to surfaces and retain activity over extended periods of time. To test this hypothesis, sterile slides were coated with the GS-2 and thymol formulation (Figure 5A) and left uncovered on a benchtop for 0, 15, 30, and 60 days before inoculating with MRSA or *P. aeruginosa*. After 1 h of exposure to GS-2 with thymol, the viable bacteria were recovered, cultured, and counted. Surprisingly, the average MRSA recovered from slides treated with GS-2 and thymol after 60 days was 3.8 CFU/mL compared to the average of >460,000 CFU/mL recovered from the water control slides (Figure 5B). Similarly, the average *P. aeruginosa* recovered from slides treated with GS-2 and thymol after 60 days was 8.8 CFU/mL compared to the >399,000 CFU/mL recovered from untreated controls (Figure 5C). This was greater than a 99.99% reduction in recovered bacteria relative to the controls. These data suggest that GS-2 with thymol could persist on surfaces and maintain antibacterial activity for at least 60 days.

## 4. Discussion

The present study described the scientific development of an antimicrobial, moving from initial characterization to standard antimicrobial testing for efficacy and completing with the development of a novel surface disinfectant. The antimicrobial activity of fatty acids is well-known, with multiple studies examining their mechanisms of action and possible utility in controlling microbial contamination. However, a primary issue with fatty acids is very low solubility in water [44].

In this study, we sought to overcome these issues by combining capric acid with L-arginine to produce a highly water-soluble fatty acid formulation that was effective against a range of bacteria, fungi, and viruses. In real-world testing, GS-2 demonstrated surface sanitizing efficacy over a current hospital-grade disinfectant. With the addition of thymol to GS-2, a rapid, robust antimicrobial was created that was active against bacteria and viruses in vitro and retained antibacterial properties on surfaces for at least two months.

Low fatty acid solubility in water has previously limited the use of fatty acids as antimicrobials. By adding L-arginine, the solubility of capric acid could be increased to 300 mg/mL. We proposed that capric acid, when combined with L-arginine, formed an ammonium carboxylate salt and that this reaction was what allowed for the high solubility in water. We demonstrated that no secondary compounds were formed upon combining these two molecules. This was not surprising since, for an amide bond to form between arginine and capric acid, the components need to be heated above 100 °C to drive off water or require a coupling reagent [45], processes which were not performed during the compounding of GS-2 components. Furthermore, reducing the pH of GS-2 in the solution resulted in the separation of the components, further supporting the formation of a soluble salt.

The mechanism of action for antibacterial fatty acids has been defined in previous studies to be heavily dependent on three critical processes: (1) increased membrane permeability due to pore formation, leading to cell lysis; (2) attack of bacterial cell membranes via the electron transport chain, decreasing the production of ATP; and (3) inhibition of membrane enzymatic activities, such as glucosyltransferase, that inhibit nutrient uptake [46,47,48,49]. Interestingly, the results presented here showed that GS-2 was up to ten times more potent against GPCs (MRSA and *S. pyogenes*) than GNRs (*E. coli* and *K. pneumoniae*). However, the required exposure time to show activity against GPCs was much longer (24 h) compared to GNRs. This finding is unusual as it is in direct contradiction with Kabara et al. (1980), where fatty acids with more than eight carbons were found not to inhibit GNR growth [50]. It is possible that the effect against GNRs observed here was due to the chemical structure of the fatty acid in the solution and a lack of required excipients needed to solubilize the fatty acid that may have improved fatty acid activity in this study. Nonetheless, GS-2 still showed good potency against hardy environmental bacteria, as well as potential bioterrorism agent surrogates *B. anthracis* Sterne and *B. thailandensis*.

During developmental testing, increasing the concentration of GS-2 in the solution did not significantly improve the time-kill against GPCs, suggesting that the mechanism of action against GPCs was relatively independent of the concentration. This effect may be due to wall teichoic acids (WTAs) that attach to the peptidoglycan cell wall of *S. aureus*, which as a result, may require more time for fatty acids to act [51]. Although the cell walls of GNRs are considerably more complex than GPCs, WTAs are not typically found on GNR cell walls [51], which could explain why, at equivalent concentrations, GNRs were more susceptible than GPCs over shorter timespans.

The addition of thymol to GS-2 significantly improved the activity of GS-2 against GPCs. GS-2 with thymol demonstrated superior time-kill activity against *S. aureus* compared to GS-2 alone but did not have a remarkable improvement on the efficacy or time-kill effect for *E. coli* or *K. pneumoniae*. Thymol was previously shown to have bacteriostatic effects against *S. aureus* at 0.31 mg/mL (MIC) and *E. coli* at 5.0 mg/mL (MIC) when exposed for longer periods of time (18–24 h) [52]. However, in disinfectant testing procedures where exposures are relatively short, thymol at 5.0 mg/mL (in DMSO) was unable to achieve a bactericidal effect. However, when 5.0 mg/mL thymol was combined with GS-2, it was able to kill both *S. aureus* and *E. coli* in timeframes acceptable for use as a surface disinfectant. This suggests that thymol may act synergistically with GS-2 to deliver bactericidal activity after just 8 min of exposure; however, further studies are required to validate this effect and the mechanism that produces it.

Like the effect against GNRs, GS-2 with thymol only showed a modest improvement in efficacy compared with GS-2 against the SARS-CoV-2 surrogate MHV-1 (from 4.92- to 5.42-log reduction of viral titer after treatment). Doubling the concentration of GS-2 and adding thymol did not produce a clear synergistic effect against the virus but rather a modest additive effect. Whilst thymol has been described as having antiviral activity, high concentrations are required, or cells need to be pre-treated with thymol before infection to observe an effect [53,54]. Thus, it is not surprising that only a modest improvement in the antiviral activity of GS-2 was observed with the addition of thymol here. Nevertheless, GS-2 with thymol still demonstrated activity against three different viruses, including non-enveloped poliovirus, which is commonly resistant to other disinfectants. This indicated that GS-2 with thymol may be a suitable antiviral agent over a broad range of virus types.

The testing performed with GS-2 and, subsequently, GS-2 with thymol supported several possible advantages of this fatty acid formulation as a multipurpose antimicrobial sanitizer, particularly over quat-based products. Firstly, GS-2 displayed activity against *C. auris*, an emerging pathogen that is considered extremely difficult to eradicate with current disinfectant products [55]. In this study, GS-2 and GS-2 with thymol also demonstrated efficacy against *C. difficile*, *P. aeruginosa*, and poliovirus, all of which are considered problematic for quats [55,56,57]. It is also worth noting that the concentrations of GS-2 (capric acid; 15–30 mg/mL) and thymol (5–7.5 mg/mL) used in this study are comparable to the concentrations of BAC used in commercial disinfectants (1–30 mg/mL) [56], suggesting that GS-2 with thymol would be an acceptable alternative to quats.

Additionally, the results from the comparative commercial building study and the glass slide experiment suggested that GS-2 remained active on surfaces for up to 60 days. Considering the structure of the ammonium carboxylate salt formed, it was not surprising that GS-2 remained unreactive to degradation. Furthermore, given the fatty acid structure, GS-2 may be less sensitive to organic material found on high-touch-point surfaces than BAC; it has been thoroughly established that quats such as BAC lose efficacy on soiled surfaces [11,12], which may account for the higher number of viable bacteria recovered from surfaces sanitized with BAC seen here. A recent study testing GS-2 in a pork abattoir also found GS-2 to have superior antibacterial activity on surfaces compared to a quat-based disinfectant [58], further supporting its capability in a real-world setting. Together, these findings suggest that GS-2 may have broader bactericidal and virucidal activity against a range of pathogens when compared to currently available disinfectants.

Another advantage of GS-2 over other sanitizers is the safety profile; even high systemic doses of capric acid did not induce any toxicities in vivo. Like L-arginine and thymol, other fatty acids also have favorable safety profiles. Together, these results suggest utilizing amino acids to increase the water solubility of fatty acids may provide a platform for exploring, combining, and developing the antimicrobial activities of other fatty acids.

The current study presented scientific evidence regarding the development and testing of a novel antimicrobial ammonium carboxylate salt derived from a fatty acid and amino acid combination. Further evidence was presented that the addition of thymol, a 10-carbon monoterpenoid, to GS-2 enhanced the antimicrobial activity of GS-2, providing improved antimicrobial and disinfectant effects. Based on these data, GS-2 and GS-2 with thymol may have significant utility in the cleaning and disinfecting of surfaces, particularly in high-risk environments where drug-resistant pathogens pose a significant threat to human health. Finally, this platform of soluble fatty acids provides novel opportunities for the development of effective and safe broad-spectrum environmental antimicrobials.

## 5. Patents

US11154524B2, 2021-10-26; Therapeutic compositions of decanoic acid and arginine. Worldwide applications CN, CA, EP, KR, BR, US, AU, JP, WO.

## Figures and Tables

**Figure 1 biomolecules-12-01293-f001:**
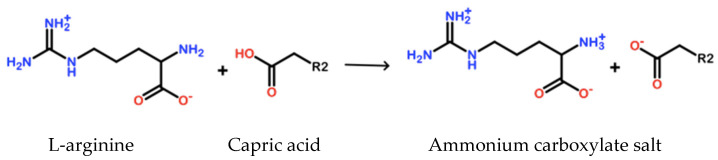
Chemical reaction of GS-2 components. An amine group on L-arginine reacts with the carboxylic acid group on capric acid to form an ammonium carboxylate salt.

**Figure 2 biomolecules-12-01293-f002:**
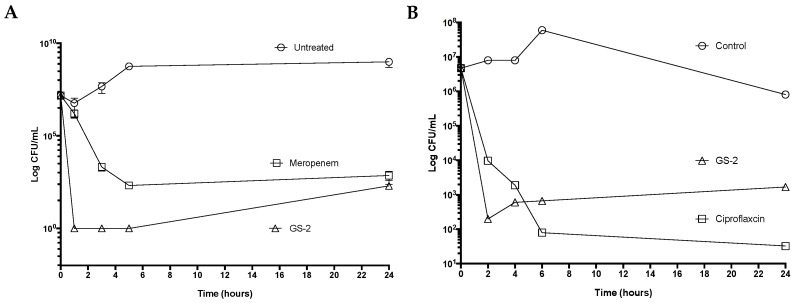
GS-2 exhibits potent antibacterial activity against high-risk bioterrorism pathogens. (**A**) *B. thailandensis* (1 × 10^6^ CFU/mL) was treated with GS-2 at 4.76 mg/mL, meropenem at 0.1 mg/mL, or untreated, and CFU/mL was determined at 0 h, 1 h, 3 h, 5 h, and 24 h. (**B**) *B. anthracis* vegetative Sterne cells (1 × 10^6^ CFU/mL) were treated with GS-2 at 4.76 mg/mL, ciprofloxacin at 0.01 mg/mL, or untreated, and CFU/mL was determined at 0 h, 2 h, 4 h, 6 h, and 24 h.

**Figure 3 biomolecules-12-01293-f003:**
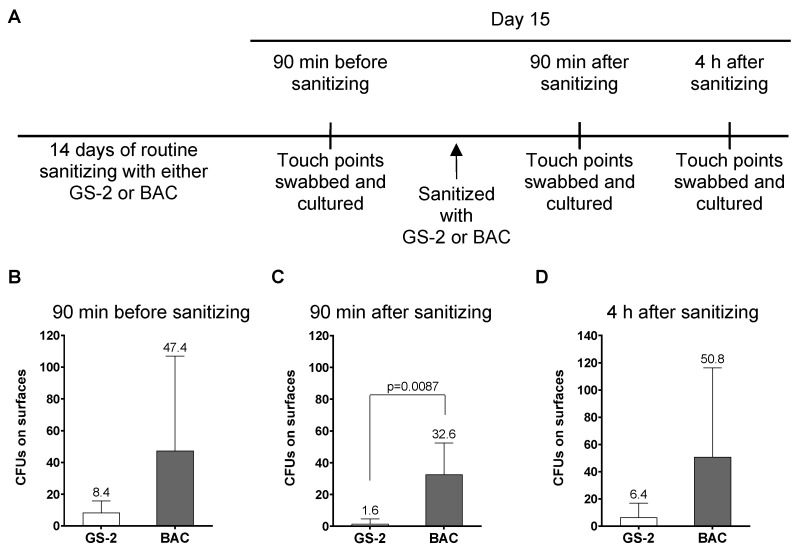
GS-2 compared to 0.42% BAC as a surface sanitizer. (**A**) Two commercial high-rise buildings were sanitized for 14 days with either GS-2 at 15 mg/mL or with a hospital-grade disinfectant containing 0.42% BAC according to product label instructions. On day 15, swabs were taken from both buildings at five high-touch-point surfaces (**B**) before sanitizing, (**C**) 90 min after sanitizing, and (**D**) 4 h after sanitizing. Swabs were then cultured on 5% SBA plates for 24 h at 37 °C, and the number of colonies counted. (Data represent mean ± SD; C, unpaired two-tailed *t*-test).

**Figure 4 biomolecules-12-01293-f004:**
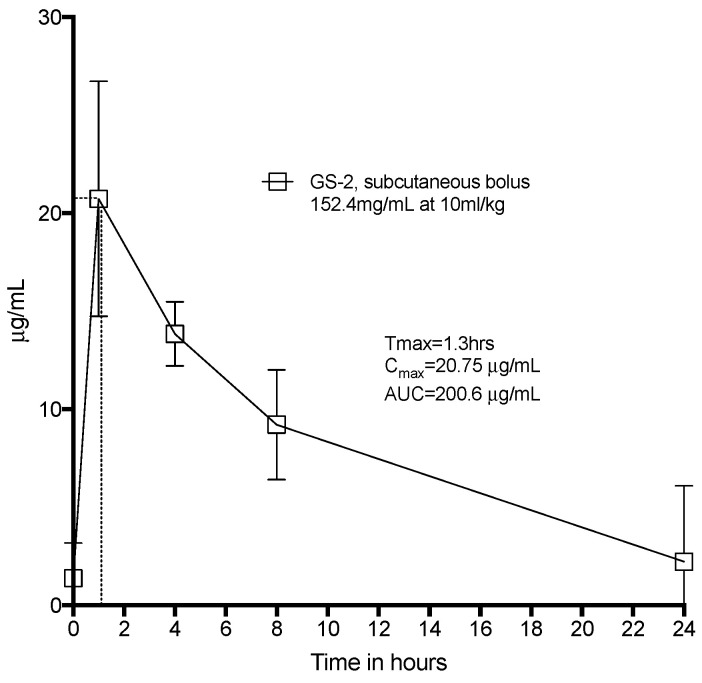
Toxicokinetics of GS-2 in Sprague-Dawley rats. Eight Sprague-Dawley rats were given a subcutaneous bolus dose of GS-2 at 10 mL/kg under the skin between the shoulder blades. Tail vein blood was drawn, processed, and assayed for capric acid at 1 h, 4 h, 8 h, and 24 h. *n* = 8, mean ± SD.

**Figure 5 biomolecules-12-01293-f005:**
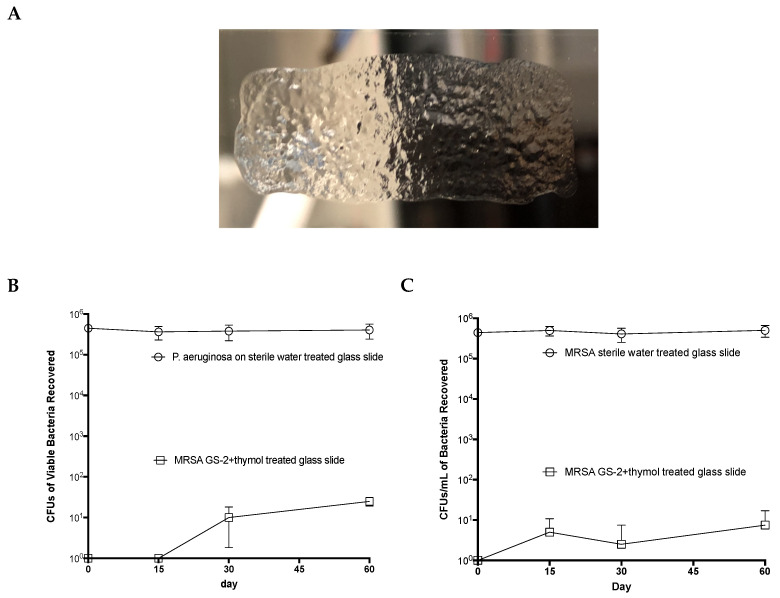
GS-2 with thymol maintains its antibacterial efficacy on surfaces 60 days after application. Sterile glass slides were coated with 200 µL of GS-2 with thymol or 200 µL of sterile water and incubated, uncovered, at 20–23 °C for up to 60 days. (**A**) Representative image of GS-2 dried on a slide. (**B**,**C**) After 0, 15, 30, or 60 days, slides were inoculated with 1 × 10^6^ CFUs of either (**B**) MRSA or (**C**) *P. aeruginosa* and incubated for 60 min at room temperature. Viable bacteria were then extracted from the slides, serially diluted, and plated onto 5% SBA plates, and colonies were counted the following day. (*n* = 4; mean ± SD).

**Table 1 biomolecules-12-01293-t001:** Testing of GS-2 against bacterial and fungal pathogens using CLSI M26-A guidelines for 24 h. *n* = 6.

Species	MIC mg/mL	MBC/MFC mg/mL
*S. aureus*	≤0.47	0.94
*B. atropheaus*	0.59	1.17
*C. butyricum*	0.29	0.59
*C. difficile* (vegetative)	-	0.15
*C. difficile* (endospores)	-	0.15
*E. coli*	3.75	3.75
*K. pneumoniae*	7.50	7.50
*P. aeruginosa*	3.75	7.50
*C. auris*	-	0.15

**Table 2 biomolecules-12-01293-t002:** Testing GS-2 against clinically isolated pathogens using CLSI M26-A guidelines for 24 h of exposure. *n* = 6.

Species	MBC (mg/mL)
MRSA	1.25
*S. pyogenes*	0.78
*E. coli*	7.04
*K. pneumoniae*	8.04

**Table 3 biomolecules-12-01293-t003:** Testing of 15 mg/mL GS-2 using TGA Option C methods for 8 min of exposure. *n* = 5 technical replicates. A pass result required at least 2 of the 5 tubes to be clear (i.e., no growth) after 48 h.

Species	Result
MRSA	Growth at 48 h, Fail
*E. coli*	No growth at 48 h, Pass

**Table 4 biomolecules-12-01293-t004:** Viral testing of 15 mg/mL GS-2 against COVID-19 surrogate using TMCV 006 ASTM E1053 methods for 10 min of exposure time; *n* = 3.

Organism	Result
MHV-1	4.92-log reduction

**Table 5 biomolecules-12-01293-t005:** Testing of GS-2 (30 mg/mL) with thymol (5 mg/mL) using TGA Option C disinfectant testing methods (8 min exposure) in 5 technical replicates. A pass result required at least 2 of the 5 tubes to be clear (i.e., no growth) after 48 h.

Agent	Organism	Result
DMSO control	MRSA	Growth at 48 h, Fail
	*E. coli*	Growth at 48 h, Fail
Thymol (5 mg/mL) in DMSO	MRSA	Growth at 48 h, Fail
	*E. coli*	Growth at 48 h, Fail
GS-2 (30 mg/mL) in DMSO	MRSA	Growth at 48 h, Fail
	*E. coli*	Growth at 48 h, Fail
GS-2 (30 mg/mL) with thymol (5 mg/mL)	MRSA	No Growth at 48 h, Pass
	*E. coli*	No Growth at 48 h, Pass

**Table 6 biomolecules-12-01293-t006:** Testing of GS-2 with thymol using CLSI M26-A guidelines for 24 h of exposure; *n* = 6. Concentrations represent concentration of capric acid in GS-2.

Species	MIC mg/mL	MBC mg/mL
*S. aureus*	≤0.55	0.55
*E. coli*	1.09	1.09
*K. pneumoniae*	2.19	4.38
*P. aeruginosa*	1.09	1.09

**Table 7 biomolecules-12-01293-t007:** Testing of GS-2 (30 mg/mL) with thymol 5 (mg/mL) for surface viricidal activity according to the TMCV 006 ATSM E1053 method for 10 min of exposure; *n* = 3.

Virus	Log_10_ Reduction of Virus
HSV-1	>6.5
Poliovirus	3.0
MHV-1	5.42

**Table 8 biomolecules-12-01293-t008:** Testing of the optimized 2:1 ratio (15.0 mg/mL GS-2 with 7.5 mg/mL thymol) under EN standard 13727:2012 + A2:2015; hospital-grade disinfectant with 5 min of exposure under dirty conditions. *n* = 3 technical replicates.

Organism	Result
*S. aureus*	>5 log reduction
*E. hirae*	>5 log reduction
*P. aeruginosa*	>5 log reduction

## Data Availability

Not applicable here.

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
