# Peer review of "GS-2: A Novel Broad-Spectrum Agent for Environmental Microbial Control"

_biomolecules, 2022, doi:10.3390/biom12091293_

Round 1
Reviewer 1 Report (Previous Reviewer 2)
The authors have addressed the concerns from the last iteration of this paper.
Author Response
We thank the reviewer for their kind response.
Reviewer 2 Report (Previous Reviewer 4)
Firstly, the activities of GS-2 is not good. The dose is too high. In addition, the results showed that thymol at the dose of 5 mg/ml did not inhibit the growth of bacteria. It is not possible. The each dose of GS-2 and thymol between the combination use is still high.
Author Response
- Regarding the concern of the reviewer about the dose of GS-2 being too high: GS-2 is not intended to be used as a therapeutic, and so the dose is not a concern in the same regard. Furthermore, the concentrations of GS-2 (capric acid; 15–30 mg/mL) used in this study are comparable to concentrations of BAC used in commercial disinfectants (1–30 mg/mL). In addition, there have been no toxicities associated with this concentration of capric acid in animal testing, so we feel this concentration of GS-2 is adequate for use as a surface sanitizer.
- Regarding the comment from the reviewer that they do not believe thymol did not inhibit the growth of bacteria: Our results indicate that 5 mg/ml thymol alone did not inhibit the growth of bacteria within 10 minutes of exposure. The short exposure time and high starting inoculum of bacteria (required for TGA Option C testing) may be the reason that thymol did not inhibit bacterial growth here.
- Regarding the concern of the reviewer about the dose of GS-2 and thymol: As mentioned in point 1, the concentrations of GS-2 (capric acid; 15–30 mg/mL) and thymol (5–7.5 mg/mL) used in this study are comparable to concentrations of BAC used in commercial disinfectants (1–30 mg/mL). Also, the GS-2 with thymol formulation is not intended to be used as a therapeutic, and so the dose is not a concern in the same regard.
Reviewer 3 Report (Previous Reviewer 5)
The scientific paper proposed by Alyce et al. entitled "GS-2: A Novel Broad-Spectrum Agent for Environmental 2 Microbial Control" , is of adequate quality and the scientific contribution will be of great interest to readers of Biomolecules. From a conceptual point of view, I recommend its publication.
Author Response
We thank the reviewer for their kind response and recommendation for publication.
This manuscript is a resubmission of an earlier submission. The following is a list of the peer review reports and author responses from that submission.
Round 1
Reviewer 1 Report
In this manuscript, the authors described a novel antimicrobial that is broad-spectrum and long-lasting. They developed a method using L-arginine to increase the amount of capric acid dissolved in water. The resulting product showed antibacterial and antifungal activity against a wide spectrum of pathogens. They optimized the formulation by adding a 10-carbon monoterpenoid and thymol, generating an antimicrobial agent that is potent and lasting in real-world scenarios. The experiments were well-designed and presented. However, a few points should be addressed before the publication of this paper.
- Line 97: Visual assessment seems quite prone to human factors. Could the author comment on why they did not use any optical measurements?
- Line 215-218: Could the authors provide images or data that support the claims they made about this adherent layer?
- Line 228-233: The authors used “data not shown” three times here. Please consider including such data in a figure/table (or in the supplementary information) since they are important to the claims made.
- Line 242-243: Does the MIC here equal the MBC/MFC (middle column) in Table 2?
- Table 3 seems unnecessary. The description in Line 275-279 should suffice.
- Figure 4: The authors should stick to using either GS-2 or Doxall. The authors should also provide more details in the Conflicts of Interest section since Doxall is a product they are currently selling.
Reviewer 2 Report
The paper by Mayfosh et al. describes the characterization of a new environmental control chemical with broad activity against various microbes. New antimicrobial agents with good safety features are needed. the GS-2 chemical has the potential to be used to control environmental microbes. However, the paper is missing a lot of information that is needed to assess the impact.
- No information is provided that verifies that the ammonium carboxylate salt was indeed created or the 10-carbon monoterpenoid thymol was added to the compound.
- No CLSI guidelines reference was listed in the reference section. What controls were used? What medium did you do your MIC analyses in for the bacteria? How long did you incubate the plates?
- No MIC results were shown for your compounds.
- Cytotoxicity of your compounds should minimally be done by trypan blue exclusion and the MTT assay would be preferred.
- The MBC concentrations for the gram-negative bacteria are quite high and the concentration of GS-2 applied was only two-fold over the MBC concentration. Why not use 30 mg/ml throughout rather than at the end?
- Many acronyms were not explained or were explained fruther into the paper rather than when first used.
- The first time using a species name you must spell out the entire genus name, thereafter, you can use the one letter abbreviation (e.g. lines 92-93).
- Temperatures and sources often missing.
- For Tables 4 and 5, what do pass and fail really mean. No explanation provided either in the figure legends or footnotes.
Reviewer 3 Report
The manuscript reports GS-2: A Novel Broad-Spectrum Agent for Environmental Microbial Control. Unfortunately, there are problems with the experimental protocol in this manuscript. To this reviewer, this manuscript does not pass the bar for publication in biomolecules.
-Line 41-44: the Authors stated that "common hospital pathogens such as Clostridium difficile, Acinetobacter baumannii, methicillin-resistant Staphylococcus aureus (MRSA), and vancomycin-resistant Enterococcus (VRE)". Acinetobacter baumannii is an important pathogen of nosocomial infection, but the author did not carry out the relevant determination of the bacteria later.
-Line 220-221: "After compounding and characterizing its physical properties", The authors did not provide characterization data.
-Line266-268: The author's selection of L-arginine and Capric acid as controls does not explain the problem, and the lack of the same type of approved fungicides as controls
-Line 325-327: After adding thymol, the effective bactericidal time of GS-2with thymol was prolonged, and there was no thymol in this experiment as a control.
-Line 135-142: The method of drug-liquid proportioning expressed by the author is difficult to understand and inconsistent with the following method.
-Figure 2:Units are not consistent (lines 129-130, both units are mg/mL). The reason for the different sampling time point settings between Figure 2A and Figure 2B is not stated.
-Line 50-52: "quats in rodents have been shown to affect neural tube formation, reproduction, respiration, and are genotoxic to cells as well as ecotoxic to aquatic life", The toxicity of GS-2 as a carboxylate salt has yet to be confirmed, and the authors pointed out that GS-2 can be used in hospital disinfection. There are no data to support the toxicity of this environmental microbial control agent to humans (eg respiratory cells).
-Line135-137: Virus selection is not representative.
Reviewer 4 Report
In the Table1 , I think the antibacterial activities is weak. The MBC is still too high although they have potential combinations.
In the tables and 5, we know that thymol per se has the potential antibacterial activities . The MIC is in the range of 0.25-0.5 mg/mL for E.coli and MRSA. However, the dose in the current study is about 5 mg/mL. It is too high.
Importantly, the whole study lacks the potential mechanism investigation.
Reviewer 5 Report
Alyce and co-authors have written a scientific paper entitled "GS-2: A Novel Broad-Spectrum Agent for Environmental 2 Microbial Control". The paper was written based on research they conducted. As a motive for writing this review paper, the authors cited the growing need to find new disinfectants that would be used to prevent the occurrence of nosocomial infections.
Since we know that the problem of nosocomial infections is more and more present, I believe that the author's idea should be supported. The work is very nice and systematically written.
It is necessary to correct:
- Line 42: Clostridium difficile (difficile); Acinetobacter baumanii (A.baumanii)
- Line 58: Escherichia coli (coli)
- In Method section need to be written references for every included experimental protocols
- Line 92: For strains isolated from clinical isolates, it should be written whether they are from a collection of strains (where is that collection located?) or were they obtained directly from a clinical laboratory (which one? Etical research purposes?)
- Klebsiella pneumoniae (pneumoniae)
- Line 94: Clinical and Laboratory Standards Institute (CLSI) standards should be added
- Line 94: I think it should be clearly written according to which CLSI standards
- Line 106: American Type Culture Collection (ATCC)
- Line 106: I think it should be clearly written according to which CLSI standards
- Line 107: Candida auris (auris)
- Line 108: Clostridium butiricum ( butiricum)
- Line 109: Bacillus atrophaeus (atrophaeus)
- Line 122: write strain data (producer) Bacillus anthracis (Sterne 34F2)
- Line 124: Burkholderia thailandensis ( thailandensis)
- Line 135: Murine Hepatitis Virus Strain 1 (MHV-1)
- Line 165: Pseudomonas aeruginosa ( aeruginosa); Salmonella choleraesuis (S. choleraesuis)
- Line 167: Streptococcus pneumoniae (pneumoniae); Streptococcus pyogenes (S. pyogenes)
- Line 208: HCL ->HCl
- Line 251: Burkholderia thailandensis (thailandensis)
- Line 473: 32 ->32 (8)
- Line 476: 91 S179 - S184 -> 91 (3B) 179S-184
- Line 544: Reference 27 - year not written (2014)